# Selection of the Best Set of Features for sEMG-Based Hand Gesture Recognition Applying a CNN Architecture

**DOI:** 10.3390/s22134972

**Published:** 2022-06-30

**Authors:** Jorge Arturo Sandoval-Espino, Alvaro Zamudio-Lara, José Antonio Marbán-Salgado, J. Jesús Escobedo-Alatorre, Omar Palillero-Sandoval, J. Guadalupe Velásquez-Aguilar

**Affiliations:** 1Centro de Investigación en Ingeniería y Ciencias Aplicadas (CIICAp), Universidad Autónoma del Estado de Morelos, Cuernavaca 62209, Morelos, Mexico; jorgearturo.sandoval@uaem.mx (J.A.S.-E.); azamudio@uaem.mx (A.Z.-L.); jescobedo@uaem.mx (J.J.E.-A.); omar.palillero@uaem.mx (O.P.-S.); 2Facultad de Ciencias Químicas e Ingeniería (FCQeI), Universidad Autónoma del Estado de Morelos, Cuernavaca 62209, Morelos, Mexico; jgpeva@uaem.mx

**Keywords:** sEMG, classification, convolutional neural network, gesture recognition, prosthesis

## Abstract

The classification of surface myoelectric signals (sEMG) remains a great challenge when focused on its implementation in an electromechanical hand prosthesis, due to its nonlinear and stochastic nature, as well as the great difference between models applied offline and online. In this work, the selection of the set of the features that allowed us to obtain the best results for the classification of this type of signals is presented. In order to compare the results obtained, the Nina PRO DB2 and DB3 databases were used, which contain information on 50 different movements of 40 healthy subjects and 11 amputated subjects, respectively. The sEMG of each subject was acquired through 12 channels in a bipolar configuration. To carry out the classification, a convolutional neural network (CNN) was used and a comparison of four sets of features extracted in the time domain was made, three of which have shown good performance in previous works and one more that was used for the first time to train this type of network. Set one is composed of six features in the time domain (TD1), Set two has 10 features also in the time domain (TD2) including the autoregression model (AR), the third set has two features in the time domain derived from spectral moments (TD-PSD1), and finally, a set of five features also has information on the power spectrum of the signal obtained in the time domain (TD-PSD2). The selected features in each set were organized in four different ways for the formation of the training images. The results obtained show that the set of features TD-PSD2 obtained the best performance for all cases. With the set of features and the formation of images proposed, an increase in the accuracies of the models of 8.16% and 8.56% was obtained for the DB2 and DB3 databases, respectively, compared to the current state of the art that has used these databases.

## 1. Introduction

The use of sEMG for the recognition of gestures focused on the control of a hand prosthesis is currently the most widely used method because it is a noninvasive measurement technique that is easy to implement [1]. Due to the stochastic, nonlinear and nonstationary nature of this type of signal [2], it is impractical to analyze the raw myoelectric signals, and that is why the typical procedure for the control of a prosthesis using this type of signal is: sEMG acquisition, digital processing, data segmentation, feature extraction, and finally classification [3], where we can find a large number of variables, such as the number of acquisition electrodes, the bandwidth of the applied digital filter, the appropriate size of the segmentation of each window, and above all, the set of features to be extracted and the optimal classification method. In this sense, recent works have shown promising results when using deep learning algorithms that can take the analysis of physiological signals to a more advanced level [4]. Currently, the use of CNN [5] to classify sEMG has given the best performance [6,7,8,9,10]. For this, the interpretation of the signal as an image to train a CNN can be divided into two; the first is to obtain a window in time and the image is composed of the number of channels (high) by the data in the time (width). Park et al. [6] used this type of image to generate a model trained by data from several subjects and obtained better performances than when using support vector machine (SVM). Atzori et al. [8] used the same representation of image to classify DB1, DB2 and DB3 databases from the Nina Pro project [11] using CNN and comparing it with traditional classification methods such as k-NN, SVM, Random Forest, and LDA, and they showed that the results were comparable in performance even using a very simple CNN structure, revealing a new area to explore. The second way to represent a sEMG image is the one proposed by Geng et al. [7], who introduced the concept of “instantaneous sEMG image”, which consists of obtaining the sEMG signals through a high-density array of acquisition channels spaced close to each other and thus record the electrical activity of the muscles in a specific area. Each value of each channel represents the value of each pixel of the image formed and allows the analysis of sEMG signals in time and space. For their work they evaluated the classification performance using these images in a CNN scheme implemented in three public databases, Nina Pro DB1, Nina Pro DB2, and CSL-HDEMG [12]; their results showed that there are indeed patterns within an instantaneous image with which gestures can be recognized, opening a new panorama for sEMG analysis. Du et al. [9] also used this type of image to evaluate the classification performance using a CNN structure in three databases, Nina Pro DB1, CSL-HDEMG [12], and CapgMyo, the latter generated by themselves. In addition, they proposed an adaptation scheme in the deep domain for the classification of gestures, thus increasing performance published in previous works. Otherwise, before starting to use CNN structures for pattern classification, the most common way to classify gesture patterns was to use feature extraction and train machine learning [13,14,15,16,17,18,19] and deep learning algorithms, mainly artificial neural networks (ANN) [20,21,22,23,24]. Within the most used features are those extracted in the time domain (TD) due to their good performance and because they do not require transformation and therefore do not require much computational time [13]. Returning to these beginnings, and also considering the good performances that CNNs have shown, some works have chosen to form images using the features extracted from the signal to train CNN networks. Hu et al. [10] extracted the set of TD features from Phinyomark [17] to propose a new sEMG image representation, where the width of the image is the number of features and the height is the number of rearranged channels [25]. They evaluated the performance using CNN, hybrid CNN-RNN and attention-based hybrid CNN on four databases, Nina Pro DB1, Nina Pro DB2, BioPatRec, and CapgMyo, obtaining the best performance published up to that time in all cases. Moreover, Wei et al. [26] used the same technique to generate images to evaluate the performance of eight sets of features in TD and three more derived from the Wavelet transform. They used a new multi-view CNN to evaluate the performance on several databases, among them the Nina Pro and BioPatRec, obtaining better performance than its predecessors. On the other hand, based on the idea that sEMG signals are by nature nonlinear and nonstationary, Pancholi et al. [27] proposed to use two features derived from power spectrum moments in time (TD-PSD1) to stabilize the signal and reduce the size of the training dataset, and using a CNN network they improved the performance in the Nina Pro DB1, DB2, and DB3 databases. Without a doubt, much progress has been made in the generation of models that obtain better performance and that allow their implementation in a real prosthesis to be more and more natural. We intend to contribute to generating a better model by proposing two things: Derived from the good performance obtained with the features TD-PSD1 [27], we propose to extract the set of five features derived from spectral moments in time (TD-PSD2), which Khushaba et al. [28] demonstrated decreases the variability in the classification performance by changing limb position and to use these features to generate the image set for training a CNN, where the width of the image is TD-PSD2 features and the height is the acquisition channels rearranged accordingly so that each signal has the opportunity to be adjacent to all the others, which allows the CNN to obtain all the possible correlations between the signals involved [25].A new type of image is proposed where not only the channels but also the features are reorganized, in such a way that the image has all the possible correlations between features and channels involved, the width of the image is TD-PSD2 features rearranged and the height is the channels rearranged.

This work is based on the hypothesis that by increasing the number of power spectrum features in the time domain and rearranging channels and features of the sEMG image to find more correlation patterns, the performance of the models generated for the DB2 and DB3 databases will improve. The results show that the best performance published so far was obtained for both databases.

## 2. Materials and Methods

In this work, the typical procedure for classifying gestures using sEMG signals was followed, adding one more step to form the images proposed for training a CNN. This section describes each of the steps shown in Figure 1.

### 2.1. sEMG Acquisition

#### 2.1.1. Nina Pro Database

The largest and most complete database with information on sEMG signals is that of the Nina Pro project. We decided to work with two of its sub-databases denoted DB2 and DB3 [29]. They used 12 electrodes for the acquisition of the signals for both databases with a sampling rate of 2 kHz, eight of the electrodes were equally spaced around the forearm at the height of the junction of the radius and humerus. Two more electrodes were placed on points of more significant activity of the fingers’ flexor muscles and extensor muscles, and two more electrodes on the points of most significant activity of the biceps and triceps. DB2 database contains data obtained from 40 intact subjects (28 males, 12 females; 34 right-handed, 6 left-handed; age 29.9 ± 3.9 years). In comparison, DB3 contains data obtained from 11 trans-radial amputated subjects (11 males; 10 right-handed, 1 left-handed; age 42.36 ± 11.96 years), the movements that were recorded for both databases are divided into three different categories, including 23 grasping and functional movements, nine wrist movements, eight hand postures, and nine finger force patterns, giving a total of 49 movements plus rest, for this work, rest was considered as one more movement giving a total of 50 movements to classify. The set of movements was selected to help cover the majority of hand motions found in activities of daily living, taking into account the taxonomy of the hand and information from robotics and rehabilitation [29]. Table 1 shows more details of both databases.

#### 2.1.2. Acquisition Protocol 

For the recording of the signals, they asked the people to perform six trials of each movement with a duration of five seconds each, alternated by three seconds of rest to avoid fatigue. The sequence in which the movements were performed was not random; that is, the six repetitions of each movement of the first category were performed, continuing with the second category and finally the third, to encourage almost unconscious repetitive movements. For acquisition, intact subjects were asked to perform the protocol with their right hand. In contrast, amputee subjects were asked to think of repeating the movements as naturally as possible with their limb missing. 

### 2.2. Signal Processing

Prior to the publication of the database, these were synchronized using high-resolution timestamps. In addition, they used a relabeling algorithm [30] to correct the errors in the synchronization of each movement made and make it coincide as best as possible with the signal that contains the information of interest. The electrodes have a 20–420 Hz Butterworth-type bandpass filter covering the frequencies of interest for sEMG signals [31]. When acquiring the signals, they also implemented a 50 Hz filter to eliminate any noise signal originating from the network power supply. For this work, we implement a 10th order bandpass digital filter in the 20–450 Hz range to guarantee the removal of any signals that are not of interest.

### 2.3. Data Segmentation

Data segmentation was performed using the overlapping windows technique [32]. We chose to use two different sizes to compare with similar works, 200 ms [10,11,28] with an overlap of 100 ms, and 150 ms [8,27] with an overlap of 25 ms, both response times less than 300 ms in order to satisfy the restrictions for its implementation in a real-time system [32] oriented to the control of an electromechanical prosthesis.

### 2.4. Feature Extraction 

We decided to use four sets of features obtained in TD. The TD1 set [33] is composed of Integrated EMG (IEMG), Variance (VAR), Willison Amplitude (WAMP), Waveform Length (WL), Slope Sign Change (SSC), and Zero Crossing (ZC), the TD2 set [34] is composed of Mean Absolute Value (MAV), SSC, WL, VAR, WAMP, ZC and four coefficients of the autoregression model (AR), these first two sets were selected due to the good performance obtained by Wei et al. [26] to classify the DB2 and DB3 databases, the third set is TD-PSD1, which consists of two features derived from the moments of the power spectrum in the time proposed by Pancholi et al. [27], the first feature is the number of peaks multiplied by the signal power (MPP) and the second feature is the zero crossings multiplied by the signal power (MZP), and the fourth set used is the TD-PSD2 features taken from Khushaba et al. [28], which have shown to reduce the variability of the error in the classification to the modify the position of the limb, and that for the first time are used for the training of a CNN network, these features were obtained from Parseval’s theorem, which establishes that the sum of the square of a function is equal to the sum of square of its transform, and is given by
(1)$$\overset{N-1}{\underset{j=0}{\sum }}{\left|x\left[j\right]\right|}^{2}=\frac{1}{N}\overset{N-1}{\underset{k=0}{\sum }}\left|X\left[k\right]X^{*}\left[k\right]\right|=\overset{N-1}{\underset{k=0}{\sum }}P\left[k\right]$$

Furthermore, this is equal to the sum of the amplitudes of the power spectrum *P[k]*, where *k* is the frequency index, *X*[*k*] is the sEMG signal expressed as a function of frequency, and *X**[*k*] is its conjugate obtained through the Discrete Fourier Transform (DFT). We must also consider that the complete description of the Fourier transform is symmetric with respect to the zero frequency and that due to this and to the fact that from the time domain, we cannot access the power spectral density, the analysis in the time domain must include the entire spectrum, considering the positive and negative frequencies, consequently, according to the definition of a moment m of order n of the spectrum *P*[*k*] which is given for
(2)$$m_{n}=\overset{N-1}{\underset{k=0}{\sum }}k^{n}P\left[k\right]$$

We can define the odd moments as zero from a statistical approach to the frequency distribution form. So, with Equation (2), we can use Parseval’s theorem to calculate the moment $m_{0}$ as
(3)$$m_{0}=\overset{N-1}{\underset{k=0}{\sum }}k^{0}P\left[k\right]=\overset{N-1}{\underset{j=0}{\sum }}{\left|x\left[j\right]\right|}^{2}$$ which is an indicator of the total power in the frequency domain, and to calculate the rest of the moments, we can use the time-differentiation property of the Fourier transform, which says that the nth derivative of a function in TD ($\Delta^{n}$) for discrete-time signals, it is equivalent to multiplying the spectrum by *k* raised to the nth power.
(4)$$F\left[\Delta^{n}x\left[j\right]\right]=k^{n}X\left[k\right]$$

Considering this property, the moment $m_{2}$ can be obtained by
(5)$$m_{2}=\overset{N-1}{\underset{k=0}{\sum }}k^{2}P\left[k\right]=\overset{N-1}{\underset{k=0}{\sum }}{\left(k^{n}X\left[k\right]\right)}^{2}=\overset{N-1}{\underset{j=0}{\sum }}{\left(\Delta x\left[j\right]\right)}^{2}$$

In the same way the moment $m_{4}$ can be calculated by
(6)$$m_{4}=\overset{N-1}{\underset{k=0}{\sum }}k^{4}P\left[k\right]=\overset{N-1}{\underset{j=0}{\sum }}{\left(\Delta^{2}x\left[j\right]\right)}^{2}$$

Then, the TD-PSD2 features are derived from calculating moments$\;m_{0}$, $m_{2}$ and$\;m_{4}$, and are detailed in Table 2. As in [28] we decided to scale logarithmically the obtained features and normalize them to obtain invariance in the scale. 

### 2.5. Image Formation 

Inspired by the good results obtained in [10], we decided to use four different methods of representing the images derived from the feature extraction of the previous section, each type of image used is described below. 

**Feature Image** is obtained directly from the feature extraction of each window, with a size of 12 × W, where 12 is the height of the image (channels) and W is the width of the image, equal to the number of features extracted, which depends on the feature set used.**MixChannel Image** is obtained by applying the rearranged algorithm to the acquisition channels as in [25], leaving an image of 72 × W, where 72 is the height of the image after applying the algorithm and W is the width of the image, equal to the number of features extracted, which depends on the feature set used.**MixFeature Image** is obtained by applying the rearranged algorithm [25] to the features, with a size of 12 × W, where 12 is the height of the image (channels) and W is the width of the image, the result of applying the algorithm to the features, leaving a different image width for each proposed set of features. For the TD-PSD1 set, this type of image is not implemented because only two features are already adjacent to each other.**Mix Image** is obtained by applying the rearranged algorithm [25] both to the channels and to the features, with a size of 72 × W, where 72 is the height of the image after applying the algorithm to the channels, and W is the width of the image, the result of applying the algorithm to the features, leaving a different image width for each set of features proposed, in the same way. For the TD-PSD1 set, this type of image is not implemented since it has only two features, and the image would be identical to the MixChannel Image.

### 2.6. CNN Architecture 

The architecture used (Figure 2) for model training is GengNet [7]. The network is composed of eight layers in total. The first two are convolutional layers with 64 filters of 3 × 3. After each of these layers, a 2 × 2 max-pooling was applied, and the subsequent two layers are locally connected with 64 filters of 1 × 1 each; for these first four layers, Batch normalization [35] was applied to reduce the internal covariance change, the subsequent three layers are fully connected with 512, 512 and 128 neurons respectively, and at the end of the network, an output layer with 50 neurons determined by the number of gestures to classify, for the first seven layers the rectified linear unit (ReLU) function was used while for the last layer a softmax function was used. For the training of the network, the stochastic gradient descent was used with a learning rate of 0.01 and a momentum of 0.9. The batch size was set to 128 and a number of epochs for training was set to 32. These hyper-parameters were selected by manual hyper-parameter tuning [36].

For the network training, we used the same scheme as in [8,10,37], which consists of using 2/3 of the repetitions of each subject as the training set; the remaining part was used as the test set; therefore, of the six repetitions of each movement detailed in Section 2.1.2, four were taken for training and two for the test. The classification accuracy for each of the combinations shown in Figure 3 was calculated as
(12)$$\mathrm{Classification}\;\mathrm{Accuracy}\;\left(\%\right)=\frac{\mathrm{Number}\;\mathrm{of}\;\mathrm{correct}\;\mathrm{classifications}}{\mathrm{Total}\;\mathrm{number}\;\mathrm{of}\;\mathrm{test}\;\mathrm{samples}}*100\%$$

## 3. Results

### 3.1. DB2 Database

The performances obtained for DB2 databases using the four sets of features and the four types of images described can be seen in Figure 4. For a window segmentation of 200 ms with an overlap of 100 ms, TD-PSD2 features showed the highest performance for all cases; however, the highest average performance for the 40 subjects was obtained when using the type of image “Mix Image” and was 87.56 ± 4.46%. The second-best performance was 87.19 ± 4.53%, and it was also obtained when using the same features and the type of image “MixChannel Image”, only 0.37% below the highest. On the other hand, the lowest performance was 75.08 ± 6.47% and was obtained with TD-PSD1 features and the type of image “Feature Image”.

The performances for the same database with a window segmentation of 150 ms with an overlap of 25 ms showed, in general, an increase in all the performances compared to the first type of segmentation; the highest average performance was also obtained with TD-PSD2 features and the type of image “Mix Image” and was 97.61 ± 1.55%. However, the second-best performance was 97.44 ± 1.11%, and this time it was obtained with TD2 features and the same type of image, only 0.17% below the highest. The lowest performance was 87.60 ± 5.52% and was obtained with the TD-PSD1 features and the “Feature Image” image type. 

The set of features that offers the best performance is TD-PSD2. For most cases, the type of image with the best performance is “Mix Image”. However, the difference in the performance obtained with the type of image “MixChannel Image” is small, and that is why we decided to make a direct comparison between the 40 subjects of the DB2 database using TD-PSD2 features, both types of image, and the two segmentation sizes. The results can be seen in Figure 5.

The results show that although there is little difference in the average performances for all subjects, in using a 200 ms segmentation the performance for 30 of the 40 subjects was increased when using “Mix Image” compared to “Mix Channel Image”, while for segmentation of 150 ms, 34 of the 40 subjects presented a better performance using “Mix Image” too.

### 3.2. DB3 Database 

The average performance obtained from the DB3 database showed similar behavior to that obtained with the DB2 database, as shown in Figure 6. The results show that when using 200 ms segmentation with 100 ms overlap, the highest performance was 74.24 ± 9.45% and was obtained when using TD-PSD2 functions with the “Mix Image” image type. The second-best performance was obtained with the same set of features but with the image type “Mix Channel Image,” and it was 73.55 ± 9.08%, only a 0.69% difference from the first one. The lowest performance was obtained with TD-PSD1 features and “Feature Image” image type, and it was 60.94 ± 9.44%. 

The results obtained for the same database with a segmentation of 150 ms with an overlap of 25 ms, as with the DB2 database, showed an increase in performance for all cases. The highest performance was 90.23 ± 6.82% and was obtained again when using TD-PSD2 features with the image type “Mix Image”, the second highest performance was obtained with the same set of features but with the image type “MixChannel Image” and was 90.03 ± 7.57%, only 0.2% below the highest, and the lowest performance was 76.77 ± 9.07% and, like the previous analyses, it was obtained with TD-PSD1 features with the image type “Feature Image”. 

The difference between the best performances, as with the DB2 database, was slight. For that reason, we decided to compare performances for each subject individually using the two schemes that gave the best results, using TD-PSD2 features and the types of images “Mix Image” and “MixChannel Image”; the results are shown in Figure 7. 

The results show that despite the little difference between the average accuracies, using segmentation of 200 ms, the performance increased for 9 of the 11 subjects when using the type of image “Mix Image,” while for segmentation of 150 ms, it was 7 of the 11 who presented an increase in performance using the same type of image.

Table 3 shows a summary of the highest classification accuracies obtained from each of the databases for both segmentations, indicating the type of image and the set of features with which they were obtained.

### 3.3. Processing Time Comparison 

For this work, an analysis of processing time was also carried out, including the time it takes for a sample to be processed from the window segmentation, the extraction of the features, and even the application of CNN model generated after training. We took five subjects from each of the databases for the analysis, and the processing time of all their samples for each of the schemes under analysis was averaged. The analysis was performed in the MATLAB R2021a software on a CPU with an Intel Core i7-6700HQ 2.60 GHz processor.

The results shown in Figure 8 are the processing times relative to the time obtained with TD-PSD2 features and the type of image “Mix Image”, which was 12 ms. This scheme was selected as the basis because it was the one that showed the best performance for both databases and both types of window segmentation.

The analysis shows that the processing time for each of the schemes is consistent with the number of features in each case and, therefore, with the size of the input images (see Table 4). For example, the TD-PSD1 set has only two features, and it is the scheme that, on average, took less time to process the samples for the “Feature Image” and “MixChannel Image” with 36.44% and 56.07% of the time that was obtained for the best performance, respectively, considering that for the other two types of images it was not implemented because mixing two features would give us the same result as the “MixChannel Image” image type. The TD2 set has the highest number of features with 10, and it is the one that, on average, took the longest processing time for all cases. With these features and the type of image “Mix Image”, the most significant difference was found, 313.76% larger processing time than the scheme with the best performance. It is essential to mention that all the calculated times are below 200 ms, the minimum response time to satisfy the restrictions of human–computer interaction [38].

The results for the DB2 database using a 200 ms segmentation showed that the highest accuracy obtained was only 0.37% higher than the second-best. On the other hand, the difference between both times was 23.08%, requiring more processing time the first one. For the case of the same database but with 150 ms segmentation, the difference in the accuracy of the scheme with better performance with TD-PSD2 features and the second with TD2 features was only 0.17%, the first being better. However, for this chance, the difference in time was more significant, with the scheme in second place taking 313.76% longer. With this analysis, the advantage of using the TD-PSD2 features scheme is accentuated because of the slight increase in precision and significant savings in processing time.

### 3.4. Comparison of Results with Previous Works 

Table 5 compares the most recent works that have used DB2 and DB3 databases, showing the highest accuracies obtained, the type of classifier used, the window segmentation size, the number of moves to classify, and the type of features extracted to get the best performance. 

## 4. Discussion

In this work, we use for the first time the type of features TD-PSD2 for the training of a CNN for pattern recognition using sEMG signals; a new type of image is also proposed where both channels and features are reorganized to search for patterns adjacent to each of them. The results show that the classification of the DB3 database has been more complicated, which was an expected result since the muscular structure of the amputee person’s limb after amputation is different from that of an intact subject [18]. From Atzori et al. [8], who obtained 75.27% and 46.27% performance for DB2 and DB3, respectively, using a 150 ms window to present, the performance in the classification of these databases has been increasing, being the CNN classifier and its variants, which have shown the best performance. The schemes that showed the highest performance in this experimental process were the set of features proposed TD-PSD2 and the type of image proposed “Mix Image”, so they are the ones that we compare with the rest of the works in Table 3. For the DB2 database with a window of 150 ms, the highest performance was 97.61%, which is 8.16% more than the highest performance achieved by [27] using the same window segmentation.

On the other hand, the scheme proposed by [27] is replicated in this work with the type of features TD-PSD1 and the type of image “MixChannel Image”, our result was 0.15% higher, that is, practically the same. For the DB2 database with a 200 ms segmentation, our best performance was 87.56%, which is 3.86% higher than the best performance obtained by [26], who used the same window segmentation and a multi-view CNN classifier. For the DB3 database with the segmentation of 150 ms, the highest performance obtained was 90.23%, 8.56% higher than that reported by [27]. When comparing similar schemes, our performance was 0.05% higher than obtained by [27], so we can say that the results were successfully replicated. For the DB3 database and segmentation of 200 ms, the highest performance obtained was 74.24%, which is 9.94% higher than the performance obtained by [26]. It should be noted that [37] reported a performance of 73.31% for the DB3 database using a 200 ms segmentation; however, they only used 10 of the 50 available moves. Even so, our performance was 0.93% higher using all moves.

In general, the comparison of our proposal with previous works shows that the features that currently give the best performance for all cases are the features in the time domain, specifically those extracted from the power spectrum in the time domain. We demonstrated that both the set of features and the type of image proposed provide an increase in the performance of the databases analyzed in this study. It is also important to mention that the performance using the 150 ms window seems to have a better result. However, the increase in performance seems to be more related to the decrease in the overlap of these windows. That is, for both [26] and in this work, an overlap of 25 ms was considered for the 150 ms windows, and they are the ones that have shown better performance so far in the literature. A direct comparison of the overlap was not made because not all the works report the size of the overlap. 

## 5. Conclusions

The most current works for gesture recognition using sEMG have obtained the best performance using CNN and feature extraction to form the training images. However, the processing time is also a factor to consider because the purpose will be to implement the model obtained in a real-time system to control an electromechanical hand prosthesis. This work focused on comparing four sets of features selected due to performance shown in previous works, one of them used for the first time to train a CNN, and with them to form four different types of images to compare performance in classification and times of processing. The conclusions of this work are:The results shown by the features obtained from the power spectrum in the time domain were the ones that showed the best performances. Additionally, when reorganizing channels and features, the performance of the model is increased.As mentioned above, the performance increases (by less than 1%) when using images with rearranged channels and features. However, the processing time for this type of image increases by approximately 20% compared to using images where only channels are rearranged.

It is important to mention that when a real-time implementation is required, the processing time should not be less than the size of a window, but the size of the overlap of that window, since it will be the time in which the system must return the result of classification. That is why, although in this and other works better performances have been obtained using a slight overlap, it should be considered that for a real implementation the processing times used by the system will be the ones that set the standard for the model to be used, that is, we can choose to use a small set of features and a small image type sacrificing the general performance of the model and use the majority voting technique to increase performance sacrificing response time, which will allow the hardware requirements not be so strict, or we can use the features and image size that give better performance and use better-capacity hardware. The time analysis performed in this study gives a perspective of the time difference involved in using one set of features or another, combined with one type of image or another, leaving the possibility of selecting a suitable scheme according to the kind of implementation to perform or to choose the appropriate hardware according to the type of scheme to be implemented.

## Figures and Tables

**Figure 1 sensors-22-04972-f001:**
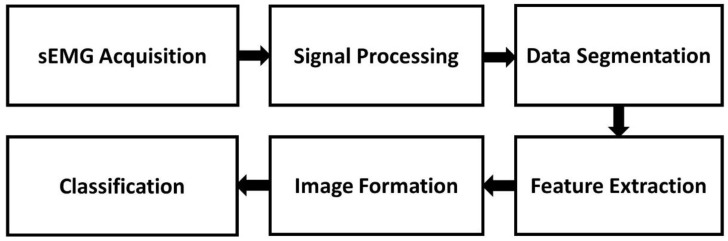
Block diagram for the classification of sEMG signals using CNN.

**Figure 2 sensors-22-04972-f002:**
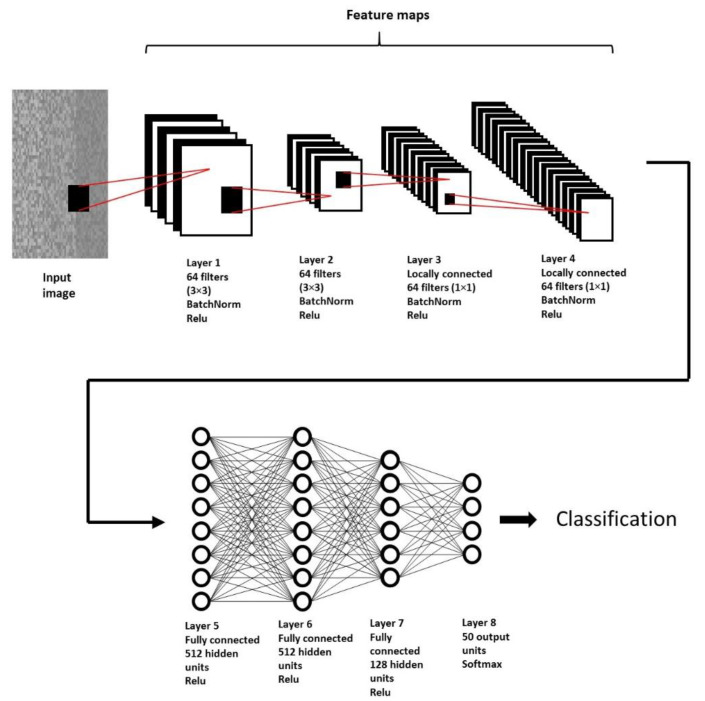
GenNet architecture composed of eight layers, this structure was used for the classification of 50 movements of the DB2 and DB3 database, using as input the images described in Section 2.5.

**Figure 3 sensors-22-04972-f003:**
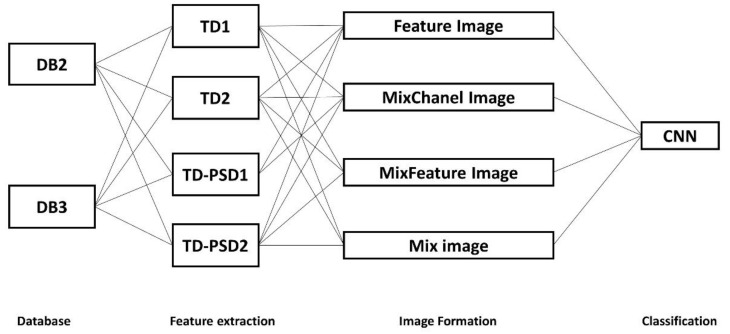
Combinations of all schemes tested for this study, using two databases, four feature extraction sets, four ways of organizing input images, and the CNN structure for classification.

**Figure 4 sensors-22-04972-f004:**
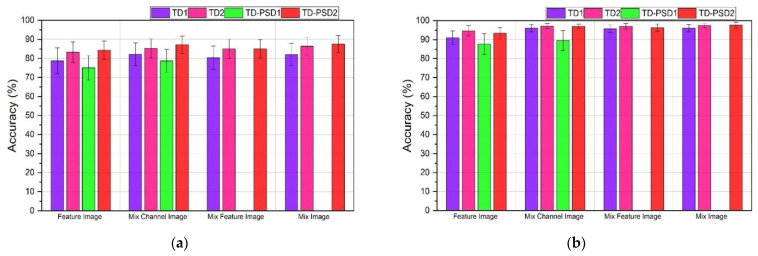
Classification accuracy using Feature Image, MixChannel Image, MixFeature Image, and Mix Image images to train a CNN network for: (**a**) DB2 database with a window size of 200 ms and overlap of 100 ms; (**b**) DB2 database with a window size of 150 ms and overlap of 25 ms.

**Figure 5 sensors-22-04972-f005:**
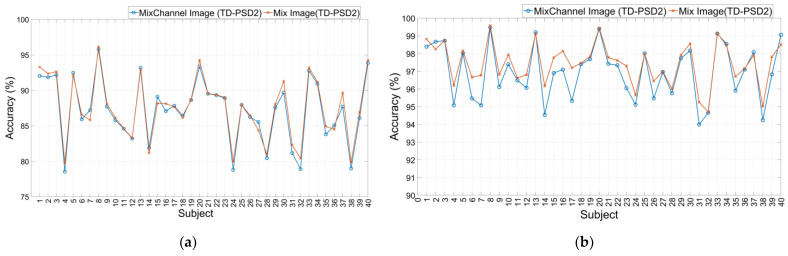
Classification accuracy of the 40 subjects of the DB2 database using: (**a**) A window size of 200 ms and overlap of 100 ms; (**b**) A window size of 150 ms and overlap of 25 ms.

**Figure 6 sensors-22-04972-f006:**
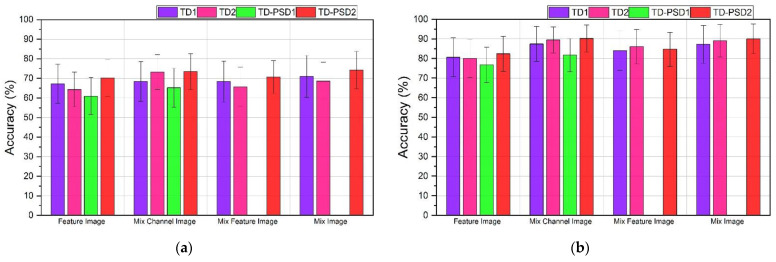
Classification accuracy using Feature Image, MixChannel Image, MixFeature Image, and Mix Image images to train a CNN network for: (**a**) DB3 database with a window size of 200 ms and overlap of 100 ms; (**b**) DB3 database with a window size of 150 ms and overlap of 25 ms.

**Figure 7 sensors-22-04972-f007:**
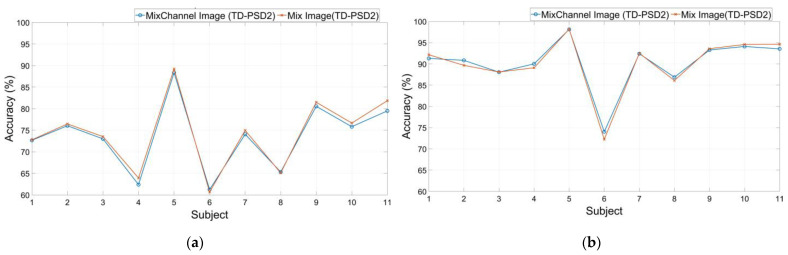
Classification accuracy of the 11 trans-radial amputated subjects of the DB3 database using: (**a**) A window size of 200 ms and overlap of 100 ms; (**b**) A window size of 150 ms and overlap of 25 ms.

**Figure 8 sensors-22-04972-f008:**
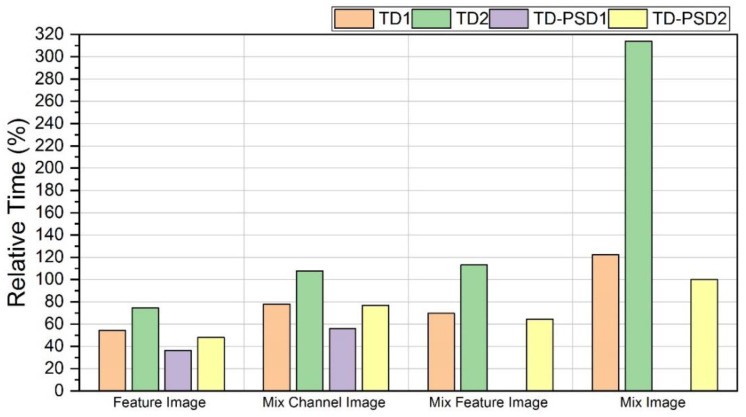
Processing time for classifying a sample relative to processing time using TD-PSD2 features with “Mix Image” image.

**Table 1 sensors-22-04972-t001:** Specifications of the databases Nina Pro used in this process experimental.

	DB2	DB3
Intact subjects	40	0
Amputated subjects	0	11
sEMG Electrodes	12 Delsys	12 Delsys
Number of gestures to be classified	50	50
Number of trials	6	6
Sampling rate	2 kHz	2 kHz

**Table 2 sensors-22-04972-t002:** Description of the five TD-PSD2 features.

Feature	Description	Equation	
$f_{1}$	Indicator of total power in the frequency domain	$\log\left(m_{0}\right)$	(7)
$f_{2}$	Noise stabilizer	$\log\left(m_{2}/m_{0}^{2}\right)$	(8)
$f_{3}$	Noise stabilizer	$\log\left(m_{4}/m_{0}^{2}\right)$	(9)
$f_{4}$	Indicator of how much energy of a vector is accumulated in few elements	$\log\left(\frac{m_{0}}{\sqrt{m_{0}-m_{2}}*\sqrt{m_{0}-m_{4}}}\right)$	(10)
$f_{5}$	Irregularity factor within a defined wavelength	$\log\left(\frac{\sqrt{\frac{m_{2}^{2}}{m_{0}*m_{4}}}}{WL}\right)$	(11)

**Table 3 sensors-22-04972-t003:** Highest classification accuracy achieved for DB2 and Db3 databases.

Database	Segmentation	Feature Set	Image Type	Classification Accuracy
DB2	200 ms	TD-PSD2	Mix Image	87.56 ± 4.46
	150 ms	TD-PSD2	Mix Image	97.61 ± 1.55
DB3	200 ms	TD-PSD2	Mix Image	74.24 ± 9.45
	150 ms	TD-PSD2	Mix Image	90.23 ± 6.82

**Table 4 sensors-22-04972-t004:** Size of each image relative to the size of the best performing image given in percentage.

Image Type	TD1	TD2	TDPSD1	TD-PSD2
Feature Image	9.0	15.1	3.0	7.5
MixChannel Image	54.5	90.9	18.1	45.5
MixFeature Image	27.2	75.7	-	16.6
Mix Image	163.6	454.5	-	100

**Table 5 sensors-22-04972-t005:** Classification accuracy of the proposed method and previous works using DB2 and DB3 databases.

Author	Database	Classes	Windows Size	Type of Features	Classifier	Accuracy in %
Atzori et al. [8] 2016	Nina Pro DB2	49	150 ms	TD	Random Forest SVM	75.27
	Nina Pro DB3					46.27
Zhai et al. [37] 2017	Nina Pro DB2	50	200 ms	Spectrogram TD	CNN	78.71
	Nina Pro DB3 *					73.31
Hu et al. [10] 2018	Nina Pro DB2	50	200 ms	TD	CNN-RNN	82.20
Wei et al. [26] 2019	Nina Pro DB2	50	150 ms		MV-CNN	82.70
			200 ms	TD **		83.70
	Nina Pro DB3		200 ms			64.30
Pancholi et al. [27] 2021	Nina Pro DB2	49	150 ms	TD PSD	DLPR	89.45
	Nina Pro DB3					81.67
This work	Nina Pro DB2	50	150 ms	TD PSD	CNN	97.61
	Nina Pro DB3		200 ms			87.56
			150 ms			90.23
			200 ms			74.24

* They only use data from 10 movement. ** They carry out the study with various types of features, in time, in frequency, and in time and frequency domain.

## Data Availability

The Nina Pro DB1 and DB2 databases were analyzed in this study. These data can be found here: http://ninapro.hevs.ch. Accessed on 20 January 2021.

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
