# Peer review of "Selection of the Best Set of Features for sEMG-Based Hand Gesture Recognition Applying a CNN Architecture"

_sensors, 2022, doi:10.3390/s22134972_

Round 1

Reviewer 1 Report

Selection of the best set of features for sEMG - based gesture 2 recognition applying a CNN architecture

Congratulations to the authors for their work.

I consider that some aspects need to be improved, which are detailed below.

Title: Make a title where there are no acronyms and where it is easy to understand and attractive at the same time. Include the type of study or research carried out.

Introduction: it is very well written, but if it were possible to shorten it a little without losing information it would be desirable.

Line:111- state your working hypothesis and null hypothesis.

Material and methods.

Line 118. Sub-section sample or subjects. Explain which subjects were selected for the study, as well as their socio-demographic characteristics.

Explain why this number of gestures was selected. Were they based on previous studies, and was the order in which the gestures were performed randomised?

Discussion.

Line 385- Put the table in the results section. And do not cite the tables in the discussion, you can talk and discuss results without citing them as they should be cited in results.

Conclusions.

Name the most important conclusions in simple sentences from your work. Do not talk about results of other works or your future lines of research, this should be mentioned in the discussion.

Reviewer 2 Report

In this paper they seek to identify from four sets of features obtained from surface myoelectric signals (sEMG) which are the best for the classification of 50 different types of movements using CNN. The sets contain time domain features. Their results show that the characteristics of power spectrum of the signal obtained in the time domain offer the best classification performance.

The authors argue that part of their novelty is to use a new type of image derived from the power spectrum of the signal features. 

I would recommend the authors to create a section of related work to demonstrate their novelty with respect to the features, since there are several papers that use time and frequency features to classify this type of signals. 

Also, they should detail a little more about the training and testing of the CNN.  How was the selection of the hyperparameters with manual tuning, how much data was used for training, how much data was used for testing?

I am left with a doubt about the numbering of the equations, since the ones in Table 2 are not numbered.
